# The Association between Time Spent in Domestic Work and Mental Health among Women and Men

**DOI:** 10.3390/ijerph20064948

**Published:** 2023-03-11

**Authors:** Anu Molarius, Alexandra Metsini

**Affiliations:** 1Centre for Clinical Research, Region Värmland, 651 85 Karlstad, Sweden; 2Department of Public Health Sciences, Karlstad University, 651 88 Karlstad, Sweden; 3Department of Knowledge Management and Patient Safety, Region Värmland, 651 82 Karlstad, Sweden; alexandra.metsini@regionvarmland.se; 4Institute of Medical Sciences, Örebro University, 701 82 Örebro, Sweden

**Keywords:** domestic work, gender, mental health, population studies, Sweden

## Abstract

Background: Unpaid domestic work has been found to be negatively associated with mental health, especially among women, in previous studies but the measures of domestic work vary. The aim of this study was to elucidate the association between time spent in domestic work and mental health in the general population. Method: The study is based on 14,184 women and men aged 30–69 years who responded to a survey questionnaire in Central Sweden in 2017 (overall response rate: 43%). Multivariate logistic regression models, adjusting for age group, educational level, family status, employment status, economic difficulties, and social support, were used to study the association between hours spent in domestic work and depressive symptoms and self-reported diagnosed depression, respectively. Results: In total, 26.7% of the respondents reported depressive symptoms and 8.8% reported diagnosed depression. No independent associations between hours spent in domestic work and depressive symptoms were found. Among women, the lowest prevalence of depression was found among those who spend 11–30 h per week in domestic work. Among men, the prevalence of self-reported diagnosed depression was highest among those who spend 0–2 h per week in domestic work, but no other statistically significant associations between time spent in domestic work and depression were found. In addition, a strong dose–response relationship was found between experiencing domestic work as burdensome and both depressive symptoms and self-reported diagnosed depression among women and men. Conclusion: Investigating time spent in unpaid domestic work may not be sufficient to assess the association between exposure to domestic work and mental health. Conversely, strain in domestic work may be a more important factor contributing to the prevalence of poor mental health in the general population.

## 1. Introduction

In general, women spend a disproportionate amount of their time carrying out three quarters of the world’s unpaid work [1]. In contrast to paid work, fewer studies have investigated the association between unpaid domestic work and mental health. Recently, several studies have raised this important but under-researched topic. The COVID-19 pandemic has made the association between domestic work and mental health more visible, especially among women, due to the increased burden of unpaid domestic work [2,3]. According to Seedat and Rondon [4], time spent on care and domestic work increased for both men and women during the COVID-19 pandemic, but the increase and intensity of this work were far greater for women. Since unpaid domestic work is associated with poor mental health [5,6], the risk of poor mental health among women engaged in unpaid work rose during the pandemic, due to exposure to greater and more stressful workloads [2,3,7]. Although most of the studies that have investigated the association between unpaid domestic work and mental health have been from high-income countries, previous studies have, for example, shown that unpaid caregivers are more anxious and depressed than non-caregivers in low- and middle-income countries [8,9]. The COVID-19 pandemic has increased the burden of domestic work even in these countries [4]. 

Recently, Ervin et al. [10] carried out a systematic review on the association between unpaid (domestic) work and mental health. They concluded that, among employed adults, unpaid labor is negatively associated with women’s mental health, with effects less apparent for men [10]. Unpaid work was measured as the amount of unpaid labor in the review. Although there was a negative association for women in most studies included in the review, some studies showed no association [11,12,13]. One of the studies included in the review was our cross-sectional population study in Sweden [14]. Ervin et al. [10] reported that, in our study, no association between spending more than 20 h per week in domestic work and self-reported diagnosed depression was found in men whereas a weak inverse association was found with depression in women. This is correct but the authors did not mention that we reported a strong positive association between experiencing domestic work as burdensome and depression in both women and men. However, the dichotomous categorization of time spent in domestic work may not have captured the true nature of the association and the use of only one measure of mental health, self-reported diagnosed depression, may have not been sufficient. We used dichotomous measures for convenience as we also wanted to assess the economic costs attributable to domestic work [14]. Therefore, and especially since different studies have produced different results, we wanted to elaborate on the previous findings with a more differentiated categorization of time spent in domestic work and to use an additional measure of mental health besides diagnosed depression. 

The aim of this study was, thus, to elucidate the association between time spent in domestic work and mental health among women and men in a general population. 

## 2. Materials and Methods

The current study is based on women and men aged 30–69 years who responded to a postal survey questionnaire sent to a random population sample during March–May 2017. The sampling frame was the population register at Statistics Sweden, the statistical administrative authority in Sweden, covering all inhabitants of the study area. The data collection was completed after two postal reminders. The aim of the survey was to monitor living conditions, lifestyle factors, and health in the general population. A total of 14,184 subjects in this age group answered the questionnaire including questions on domestic work [14]. The overall response rate was 43%. The area investigated covers 55 municipalities in five counties with more than one million inhabitants in Central Sweden. 

### 2.1. Outcomes and Major Risk Factors

Depressive symptoms were assessed with the following question: “Do you have any of the following discomforts or symptoms?”. One of the discomforts or symptoms was dejection, with the response options “No”, “Yes, minor discomfort”, and “Yes, severe discomfort”. This question is identical to the question used in the national public health survey conducted by the Public Health Agency of Sweden [15]. The latter two options were considered as having depressive symptoms. 

Self-reported diagnosed depression was assessed with the following question: “Do you have any of the following diagnosed illnesses?”, where depression was one of the illnesses listed (with answer options yes/no).

There were two questions about domestic work. The first one was: “How many hours a week on average do you spend working at home that is not paid work? E.g. taking care of children, nursing relatives, buying the groceries, cooking, paying the bills, washing the laundry, cleaning, taking care of a car, a house or a garden”. The answer options were: 0–2, 3–10, 11–20, 21–30, and 31 or more hours per week. The second question asked how often the respondent experienced domestic work as burdensome (never, seldom, sometimes, most of the time, or all the time). The alternatives were combined into three categories: “never/seldom”, “sometimes”, and “all or most of the time”.

### 2.2. Covariates

Educational level was obtained from a national education register and categorized into three levels: low (elementary school), medium (upper secondary school), and high (at least 3 years of university or corresponding education). Family status was obtained from a survey question and categorized into living alone, living with a partner, living with a partner and children, single parent, and other. Employment status was derived from a survey question about whether the respondent was employed (including self-employed), student, unemployed, or other. 

Economic difficulties were estimated with the question “During the last 12 months, have you ever had difficulty in managing the regular expenses for food, rent, bills etc.?” (“no”, “yes, once”, or “yes, more than once”). The alternatives were dichotomized into no and yes.

Social support was assessed with the question “Do you have anyone you can share your innermost feelings with and confide in?” (yes/no).

### 2.3. Ethical Considerations

The individuals in the sample were informed that responded questionnaires would be linked to the Swedish official registries through personal identification numbers, to achieve information on gender, age, and educational level. The respondents, thus, accepted the linking of registry data by informed consent. The personal identification numbers were deleted directly after the record linkage. Statistics Sweden carried out the sampling, collected the data, performed the linkage with register data, and delivered de-identified data to the county councils/regions. All procedures involving human subjects were approved by the Regional Ethics Board in Uppsala (EPN 2015/417). The data are protected according to the Public and Privacy Act (2009:400, Chapter 24, Section 8).

### 2.4. Statistical Analyses

The analyses were performed separately for men and women. The associations between hours spent in domestic work and depressive symptoms as well as self-reported diagnosed depression were analyzed using multivariate logistic regressions. Since there is a U-shaped association between hours spent in domestic work and burdensome domestic work [16], spending 3–10 h per week was used as the reference category. Age group, educational level, family status, employment status, economic difficulties, and social support were treated as potential confounders and adjusted for in the regression models. In the final model, the odds ratios were also adjusted for burdensome domestic work. The results are reported as odds ratios (OR) and 95 percent confidence intervals (95% CI) for depressive symptoms and self-reported diagnosed depression, respectively. 

## 3. Results

Women reported more hours spent on average in domestic work than men did; 10.5% of the women reported that they spend more than 30 h per week in domestic work compared to 5.6% of the men (Table 1). In addition, 9.6% of the women and 5.7% of the men experienced domestic work, all or most of the time, as burdensome. In total, 31.2% of the women and 20.9% of the men reported depressive symptoms. The corresponding prevalence of self-reported diagnosed depression was 10.5% in women and 6.5% in men.

There was a statistically significant association between time spent in domestic work and depressive symptoms in the univariate analysis in both men and women (Table 2). However, these associations disappeared after adjusting for age group, educational level, family status, employment status, economic difficulties, and social support. Further adjustment for burdensome domestic work did not affect these associations. Nevertheless, a strong association was found between burdensome domestic work and depressive symptoms among both women and men, with the highest odds ratios among those who experienced domestic work as burdensome all or most of the time.

When examining self-reported diagnosed depression, the prevalence was highest among those who spend 0–2 h per week in domestic work among both women and men before adjustment for potential confounders (Table 3). No other statistically significant associations between time spent in domestic work and depression were found among men, neither before nor after adjusting for burdensome domestic work. Among women, the difference between those who spend 0–2 h per week and those who spend 3–10 h per week was no longer statistically significant after adjusting for burdensome domestic work. The lowest prevalence of depression among women was found among those who spend 11–30 h per week in domestic work. As observed for depressive symptoms, there was a strong association between burdensome domestic work and depression among both women and men.

## 4. Discussion

In total, 26.7% of the women and men aged 30–69 years included in the current study, based on a survey carried out in 2017, reported depressive symptoms and 8.8% reported diagnosed depression. The prevalence of both measures of mental health problems was higher in women than in men. Women spent more time in domestic work than men did. No independent associations between hours spent in domestic work and depressive symptoms were found. Among women, the lowest prevalence of depression was found among those who spend 11–30 h per week in domestic work. Among men, the prevalence of self-reported diagnosed depression was highest among those who spend 0–2 h per week in domestic work but no other statistically significant differences in depression between categories of time spent in domestic work were found. 

Ervin et al. [10] concluded in their systematic review that unpaid work among employed adults is negatively associated with women’s mental health, whereas the effects are less apparent in men. Unpaid work was measured as the amount of unpaid labor in the review. Investigating time spent in unpaid labor may however not be sufficient to assess the association between exposure to unpaid labor and mental health. This has also been commented on by Ervin et al. [10] and others [11]. Different types of domestic work may differ in strain and control over tasks and thus have different implications for mental health. Low-schedule-control tasks, such as laundry and cooking, must be typically performed daily and at certain times, whereas high-schedule-control tasks, such as yard work and car maintenance, can often be done without any time urgency [11]. Women often spend more hours in low-schedule-control tasks whereas men spend more hours in high-schedule-control tasks [11]. The type of domestic work may thus contribute to the difference between men and women in the association between the number of hours spent in domestic work and mental health. Some types of domestic work may also be favorable for health if the person can choose the tasks they prefer [17]. It is, however, not often possible to choose between the tasks since it is necessary for some tasks to be carried out daily. Additionally, the review of Ervin et al. [10] confirmed that there are substantial gender differences in exposure to unpaid work and every included study reported women doing more, regardless of their geographical or temporal setting. There are, however, differences between high-income countries and lower-income countries but also between different income groups within countries [4]. Higher-earning women—and men—in all countries may be able to give more attention to and spend more time with their children by outsourcing more burdensome household tasks [4]. 

The gendered nature of unpaid work became more apparent during the COVID-19 pandemic. A study from the UK found that women spent much more time in unpaid care work than men did during the COVID-19 lockdown, and it was more likely to be the mother than the father who reduced working hours or changed employment schedules due to increased time on childcare [7]. Women who spent long hours in housework and childcare were more likely to report increased levels of psychological distress, particularly among lone mothers [7]. An Australian study showed that the higher risk of symptoms of anxiety and depression among women during the COVID-19 restrictions was, in part, explained by the disproportionate burden of unpaid caregiving among women [2]. A study from Chile found that women reported an increase in household chores and childcare and an increase in poor mental health during the pandemic [3]; a study from Canada reported an increase in anxiety and depression among mothers during the pandemic [18]. The current study was, however, based on a survey carried out before the pandemic, which may have resulted in an underestimation of the levels of domestic work [4,7] and the prevalence of mental health problems, especially among women [4,19].

In addition to mental health, long domestic working hours (>25 per week) have been found to be associated, for example, with an increased risk of medically certified sickness absences in both women and men in longitudinal studies [20]. In the present study, there were some differences in the associations between hours spent in domestic work and mental health depending on the measure of mental health. Depressive symptoms were not at all associated with time spent in domestic work, neither before nor after adjustment for burdensome domestic work, whereas some statistically significant differences in the odds ratios were found for the different categories of hours spent in domestic work for depression. Among women, the lowest prevalence of depression was found among those who spend 11–30 h per week in domestic work both before and after adjustment for burdensome domestic work. This explains the weak negative association between hours spent in domestic work and depression found among women in the previous study where the cut-off point was 20 h per week [14]. The association was, however, weaker for hours spent in domestic work than for burdensome domestic work. The association between hours spent in domestic work and mental health is not straightforward since several previous studies have reported no association [11,12,13,21]. On the other hand, some previous longitudinal studies have found that high unpaid workloads among working populations increased the risk for psychological distress [22] and depressive symptoms [6].

When examining self-reported diagnosed depression, the prevalence was highest among those who spend 0–2 h per week in domestic work among both women and men, although this association was no longer statistically significant among women when adjusted for burdensome domestic work. The high prevalence of depression among those who spend very little time in domestic work is probably mainly due to reverse causality, since persons with depression, and other health problems, may be unable to perform domestic work. This category covers, however, a very small proportion of the adult population, 4.6% among women and 9.9% among men.

A strong relationship was found between experiencing domestic work as burdensome and both depressive symptoms and self-reported diagnosed depression among women and men. We found this for depression also in the previous study [14]; however, the measure in that study was dichotomized. This time we could show that there is a dose–response relationship and that this was true for both measures of mental health. Nevertheless, experiencing domestic work as burdensome and hours spent in domestic work are related. There is a U-shaped association between hours spent in domestic work and burdensome domestic work [16]. Both those who spend very little time and those who spend very much time in domestic work experience it as burdensome.

Several studies have examined the association between inequity in the responsibility for domestic work and mental health among employed persons [11,21,23]. While this is an important question, it excludes persons living alone and single parents. In the previous study, we found that the strong association between burdensome domestic work and self-reported diagnosed depression was not restricted to employed persons or parents with children [14]. Our measure of experiencing domestic work as burdensome is subjective and perhaps suboptimal, but the strong association found with both depressive symptoms and self-reported diagnosed depression is in line with other studies showing strong negative associations between strain/demands in domestic work and mental health [24,25,26]. The results underline that there is a need to make unpaid domestic work and its association with mental health more visible to policymakers and to implement policies that provide a more equal and relieved burden of domestic work. Flexible working options, available childcare services, and subsidies for redistributing unpaid work are some examples that have been suggested to reduce the burden [4].

### Limitations

The main limitation of our study is the cross-sectional design, which prevents any interpretations of causality. It is, therefore, not possible to know whether and to what degree the high prevalence of depression among those who spend 0–2 h per week in domestic work is due to reverse causality, i.e., that persons with depression are unable to carry out domestic work. However, longitudinal studies have shown that, for example, combined work and family demands are a risk factor for poor mental health [26,27]. Most of the studies included in the review of Ervin et al. [10] were also cross-sectional in design, limiting causal inference. The results may also be affected by the measure of mental health used, as was the case in the current study. Another limitation is that the survey was carried out in 2017, before the COVID-19 pandemic. However, it is unlikely that the pandemic has significantly affected the association between domestic work and mental health, although it has likely increased the burden of domestic work.

The response rate of the survey was 43%, which may affect the representativeness of the respondents and lead to an underestimation of domestic work and mental health problems. Another limitation regards the validity of the measures of domestic work. There are, however, no validated measures to examine domestic work due to the lack of research in this field [10,11,25]. The question on depressive symptoms was identical to the question used in the national public health survey in Sweden [15]. Diagnosed depression was self-reported, and the prevalence was somewhat lower (9%) than the proportion using antidepressants (12%) in this age range in 2017 in Sweden [28]. Previous studies have however found that self-reported clinician-diagnosed depression is of adequate validity to measure depression [29,30].

One of the strengths of the current study is that it is based on a considerable random sample of the general population. The study population also included a reasonable proportion of both women and men, which is important since many of the studies on this association have only included women [22,23,24,25]. A further strength is that the study was not restricted to employed persons or parents with children.

We agree with the notion that unpaid domestic work is associated with poor mental health, especially in women, but that robust longitudinal studies are needed. The need is even more urgent due to the high economic burden of depression and other mental health problems to both the individual and society [14,31]. The heterogeneity Ervin et al. [10] found between the studies included in the review and the lack of validated measures indicated in previous studies [11,23], further emphasizes that accomplishing more standardized ways to measure different dimensions and types of unpaid labor would benefit this important research field. Further research is also needed to examine the long-term effects of the COVID-19 pandemic on domestic work in different parts of the world and the relationship between unpaid domestic work and mental health in different populations.

## 5. Conclusions

Investigating time spent in unpaid domestic work may not be sufficient to assess the association between exposure to domestic work and mental health. Conversely, strain in domestic work may be a more important factor contributing to the prevalence of poor mental health in the general population. This has bearing on designing policies to reduce the strain of domestic work in the general population, especially among women.

## Figures and Tables

**Table 1 ijerph-20-04948-t001:** Number of respondents, distribution of time spent in domestic work, burdensome domestic work, and the prevalence of depressive symptoms and self-reported diagnosed depression in women and men aged 30–69 years.

	Women	Men	Total
N	7981	6203	14,184
Time spent in domestic work (%)			
0–2 h/week	4.6	9.9	6.9
3–10 h/week	38.2	47.6	42.3
11–20 h/week	31.8	28.6	30.4
21–30 h/week	14.9	8.4	12.0
31+ h/week	10.5	5.6	8.4
Experiences domestic work as burdensome (%)			
Never/seldom	43.2	61.2	51.0
Sometimes	47.3	33.1	41.1
All or most of the time	9.6	5.7	7.9
Depressive symptoms (%)			
Yes	31.2	20.9	26.7
No	68.8	79.1	73.3
Self-reported diagnosed depression (%)			
Yes	10.5	6.5	8.8
No	89.5	93.5	91.2

**Table 2 ijerph-20-04948-t002:** Unadjusted and multivariate odds ratios (95% confidence intervals in parentheses) for depressive symptoms among women and men aged 30–69 years.

	OR ^1^ (95% CI)	OR ^2^ (95% CI)	OR ^3^ (95% CI)
**Women**			
Time spent in domestic work			
0–2 h/week	**1.5 (1.2, 1.8)**	1.2 (0.9, 1.5)	1.0 (0.8, 1.3)
3–10 h/week	1	1	1
11–20 h/week	1.0 (0.9, 1.1)	1.0 (0.8, 1.1)	0.9 (0.8, 1.0)
21–30 h/week	**1.2 (1.0, 1.4)**	1.1 (0.9, 1.3)	1.0 (0.8, 1.1)
31+ h/week	**1.3 (1.1, 2.5)**	1.0 (0.9, 1.2)	0.9 (0.8, 1.1)
Experiences domestic work as burdensome			
Never/seldom			1
Sometimes			**2.4 (2.1, 2.7)**
All or most of the time			**6.8 (5.7, 8.3)**
**Men**			
Time spent in domestic work			
0–2 h/week	**1.6 (1.3, 2.0)**	1.2 (1.0, 1.5)	1.1 (0.9, 1.4)
3–10 h/week	1	1	1
11–20 h/week	1.0 (0.8, 1.1)	1.0 (0.9, 1.2)	1.0 (0.9, 1.2)
21–30 h/week	1.1 (0.9, 1.4)	1.1 (0.8, 1.4)	1.0 (0.8, 1.3)
31+ h/week	1.1 (0.9, 1.5)	1.1 (0.8, 1.4)	1.0 (0.7, 1.3)
Experiences domestic work as burdensome			
Never/seldom			1
Sometimes			**2.6 (2.2, 3.0)**
All or most of the time			**5.7 (4.4, 7.4)**

^1^ Unadjusted odds ratio. ^2^ Adjusted for age group, educational level, family status, employment status, economic difficulties, and social support. ^3^ Adjusted for age group, educational level, family status, employment status, economic difficulties, social support, and burdensome domestic work. Statistically significant odds ratios are marked with bold.

**Table 3 ijerph-20-04948-t003:** Unadjusted and multivariate odds ratios (95% confidence intervals in parentheses) for self-reported diagnosed depression among women and men aged 30–69 years.

	OR ^1^ (95% CI)	OR ^2^ (95% CI)	OR ^3^ (95% CI)
**Women**			
Time spent in domestic work			
0–2 h/week	**1.8 (1.3, 2.4)**	**1.4 (1.0, 2.0)**	1.2 (0.8, 1.6)
3–10 h/week	1	1	1
11–20 h/week	**0.8 (0.7, 1.0)**	**0.8 (0.7, 1.0)**	**0.8 (0.7, 1.0)**
21–30 h/week	0.8 (0.6, 1.0)	**0.7 (0.5, 0.9)**	**0.6 (0.5, 0.8)**
31+ h/week	1.1 (0.9, 1.4)	0.8 (0.6, 1.1)	0.7 (0.6, 1.0)
Experiences domestic work as burdensome			
Never/seldom			1
Sometimes			**1.9 (1.6, 2.4)**
All or most of the time			**5.3 (4.2, 6.7)**
**Men**			
Time spent in domestic work			
0–2 h/week	**2.6 (1.9, 3.4)**	**1.7 (1.2, 2.3)**	**1.4 (1.0, 2.0)**
3–10 h/week	1	1	1
11–20 h/week	1.0 (0.8, 1.4)	1.2 (0.9, 1.6)	1.2 (0.9, 1.6)
21–30 h/week	1.1 (0.7, 1.7)	1.1 (0.7, 1.7)	1.0 (0.6, 1.5)
31+ h/week	1.4 (0.9, 2.2)	1.4 (0.9, 2.3)	1.4 (0.8, 2.2)
Experiences domestic work as burdensome			
Never/seldom			1
Sometimes			**3.0 (2.3, 3.9)**
All or most of the time			**7.4 (5.3, 10.4)**

^1^ Unadjusted odds ratio. ^2^ Adjusted for age group, educational level, family status, employment status, economic difficulties, and social support. ^3^ Adjusted for age group, educational level, family status, employment status, economic difficulties, social support, and burdensome domestic work. Statistically significant odds ratios are marked with bold.

## Data Availability

The dataset analyzed during the current study is not publicly available due to confidentiality and regulations under the Swedish law (the Public and Privacy Act 2009:400, Chapter 24, Section 8).

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
