# Peer review of "The Association between Time Spent in Domestic Work and Mental Health among Women and Men"

_ijerph, 2023, doi:10.3390/ijerph20064948_

Round 1
Reviewer 1 Report
The article submitted for review seeks to clarify the relationship between time spent on housework and mental health for women and men in the general population. The authors note that this is their continuation of research on the links between housework and mental health. In my opinion, the aim has been achieved and the paper fits well within the theme of the proposed journal issue. The strengths of the article are the conduct of the research on a large sample of subjects and the well-paced narrative throughout the text.
The abstract and keywords are written correctly.
The introduction is reasonably short. Already in the first paragraph the research on the COVID-19 pandemic is presented, which might suggest that the research will refer to this event. Meanwhile, it is only a brief mention, which has no connection to the current research, which was carried out in 2017. In the discussion, the authors return to this thread again in the form of some 'interjection', albeit already a bit longer, but it is still not related to the current research.
The research design is interesting and covers a large group. My only concern is the question regarding the participants' recognition of symptoms indicative of depression, which assumes that the participants become experts in diagnosing their own disorders and illnesses.
From an ethical point of view, the research does not raise any objections.
Findings
The results of the study are presented clearly, comprehensively and in accordance with the art of research. The tables are clear and transparent. All variables are included and discussed. Certainly a strength of the research is the reasonably similar number of women and men taking part in the research, which is not always obvious or easily achieved as men are less willing to complete any questionnaires.
Discussion
The discussion takes up a good portion of the article. It begins with a summary of the results of our own research. Again, there is a long paragraph on pandemics, which is not relevant to the research conducted. I do not think it is necessary. Mayby only as a suggestion for future research...
The comparison of their own research with other results is correct. The authors are able to explain the results of their own research in comparison to others.
The authors present both the strengths and weaknesses of their research. In my opinion, the weakness of the research is certainly the year it was conducted (2017). In the meantime, there have been many changes in the world, mainly related to the pandemic, which has had a huge impact on the lives of individuals and communities. Perhaps it would be valuable to repeat the study in the current year on the same research group????
The bibliographic references are in line with the theme of the article and refer to recent ones up to a period of about 20 years. Of course, most of the research on domestic work and health carried out in recent years is mainly concerned with the pandemic period and the changes that have taken place both in the labour market and in health issues at the individual and community level, hence the appearance of older publications in the article under discussion.
Author Response
Reviewer 1
The article submitted for review seeks to clarify the relationship between time spent on housework and mental health for women and men in the general population. The authors note that this is their continuation of research on the links between housework and mental health. In my opinion, the aim has been achieved and the paper fits well within the theme of the proposed journal issue. The strengths of the article are the conduct of the research on a large sample of subjects and the well-paced narrative throughout the text.
The abstract and keywords are written correctly.
The introduction is reasonably short. Already in the first paragraph the research on the COVID-19 pandemic is presented, which might suggest that the research will refer to this event. Meanwhile, it is only a brief mention, which has no connection to the current research, which was carried out in 2017. In the discussion, the authors return to this thread again in the form of some 'interjection', albeit already a bit longer, but it is still not related to the current research.
Authors reply: Thank you for pointing this out. We have now indicated more clearly that the survey the current study is based on was carried out in 2017 i.e. before the covid-19 pandemic. The pandemic was taken up in the introduction and discussion because the pandemic has made the association between unpaid domestic work and mental health more visible and several of the more recent publications investigate the consequences of the pandemic on domestic work and this association.
The research design is interesting and covers a large group. My only concern is the question regarding the participants' recognition of symptoms indicative of depression, which assumes that the participants become experts in diagnosing their own disorders and illnesses.
Authors reply: We have extended the discussion on the validity of the measures of mental health with some additional references.
From an ethical point of view, the research does not raise any objections.
Findings
The results of the study are presented clearly, comprehensively and in accordance with the art of research. The tables are clear and transparent. All variables are included and discussed. Certainly a strength of the research is the reasonably similar number of women and men taking part in the research, which is not always obvious or easily achieved as men are less willing to complete any questionnaires.
Authors reply: We have commented this in the strengths of the study.
Discussion
The discussion takes up a good portion of the article. It begins with a summary of the results of our own research. Again, there is a long paragraph on pandemics, which is not relevant to the research conducted. I do not think it is necessary. Mayby only as a suggestion for future research...
Authors reply: As mentioned above we have explained why the pandemic was brought up in the discussion and that our survey was conducted before the pandemic. We have also added that it should be considered in future research.
The comparison of their own research with other results is correct. The authors are able to explain the results of their own research in comparison to others.
The authors present both the strengths and weaknesses of their research. In my opinion, the weakness of the research is certainly the year it was conducted (2017). In the meantime, there have been many changes in the world, mainly related to the pandemic, which has had a huge impact on the lives of individuals and communities. Perhaps it would be valuable to repeat the study in the current year on the same research group????
Authors reply: We fully agree and have added this to the limitations. We have also indicated that the pandemic should not have affected, at least not to a major part, the association between domestic work and mental health, although it probably has affected the levels of domestic work and mental health problems.
The bibliographic references are in line with the theme of the article and refer to recent ones up to a period of about 20 years. Of course, most of the research on domestic work and health carried out in recent years is mainly concerned with the pandemic period and the changes that have taken place both in the labor market and in health issues at the individual and community level, hence the appearance of older publications in the article under discussion.
Authors reply: We interpret this comment that the references are ok. Please note that we have revised the introduction somewhat, in line with the comments received from reviewer 2, and added some new references as well as moved some references from discussion to introduction.
Reviewer 2 Report
Comments and suggestions
It was my pleasure to review this manuscript dealing with the association between time spent in domestic work and mental health among women and men. This manuscript aims to examine the association between time spent in domestic work and mental health in the general population. A cross-sectional study was conducted in Mid-Sweden in 2017 using a postal survey questionnaire sent to a random population sample. The overall response rate was 43%. In brief, I found the topic quite interesting but this manuscript lacked to be discussed deeply about their findings. But with the sole objective of improving the quality of the manuscript, I will allow myself to make a few comments.
Introduction part:
1. I suggest the introduction section may add some references from other countries about this topic such as different types of domestic work and mental health or inequity in responsibility for domestic work in the family and mental health. In the present manuscript, the author repeatedly addressed the association between unpaid domestic work and mental health from the results of Ervin’s study or their prior study in Sweden. It can be enriched with further references drawing from the current rich international bibliography or references, especially during the pandemic period.
Materials and Methods
1. As you mentioned in the manuscript, the relevant topic of this study has been published before. I suggest that this manuscript still address briefly how to design this study’s sampling and studying process or cite the previous study's reference to let readers understand thoroughly.
2. The author needs to explain why you adopt spending 3- 10 hours per week as the reference category in this study. Is there any study recommending adopting this cut-point to analyze? This part should mention in the revised manuscript.
Results
1. Parahraph 1, lines 124-129:
In the present manuscript, rounding the number was adopted. Please let the number of percentages accord your table 1 and avoid readers getting confused.
Discussion
1. Lines 169-170:
Rounding the number was adopted. Please let the number of percentages accord your table 1 and avoid readers getting confused.
2. Lines 173-178:
You mentioned that “ among women, the lowest prevalence of depression was found among those who spend 11-30 hours per week in domestic work. Among men, the prevalence of self-reported diagnosed depression was highest among those who spend 0-2 hours per week in domestic work but no other statistically significant differences in depression between categories of time spent in domestic work were found”. The authors should discuss or clarify here why these results appear like this and does your study have any practical and specific policy implications in this area. Please discuss your opinion and add this important information with scientific references to the discussion part.
3. Paragraph 2 (page 6):
The previous references of Ervin et al. [6] and others [7] have concluded that investigating time spent in unpaid labor may however not be sufficient to assess the association between exposure to unpaid labor and mental health. However, your study still adopted time spent to examine this objective. This leads to misunderstanding results and may lead to bias. This part should discuss and mention the study's limitations.
4. That standard instruments, such as the concept of depressive symptoms, experience domestic work as burdensome, social support, economic difficulties, and were not applied in the present study, could have led to a misclassification of information, may have affected the validity and reliability of the questionnaire, and may have caused a potential lack of comparability to the results of prior studies that used standard instruments. This part should discuss and mention the study's limitations.
5. This study has a lower response rate as well as 43%. Selection bias due to non-participants was inevitable. This part should discuss and mention the study's limitations.
6. Page 7, lines 238-239:
You mentioned that “This category covers, however, a very small proportion of the adult population; 5% among women and 10% among men”. In this manuscript, I did not see any data to demonstrate this sentence. Please confirm it or add the necessary material to the revised manuscript.
7. In this study, the experience of domestic work as burdensome was an important predictor instead of time spent hours in domestic work. The authors discussed a lot in time spent hours. I suggest the authors should focus on discussing the impact of the experience of domestic work as burdensome on mental health in the discussion part.
Author Response
Reviewer 2
It was my pleasure to review this manuscript dealing with the association between time spent in domestic work and mental health among women and men. This manuscript aims to examine the association between time spent in domestic work and mental health in the general population. A cross-sectional study was conducted in Mid-Sweden in 2017 using a postal survey questionnaire sent to a random population sample. The overall response rate was 43%. In brief, I found the topic quite interesting but this manuscript lacked to be discussed deeply about their findings. But with the sole objective of improving the quality of the manuscript, I will allow myself to make a few comments.
Introduction part:
- I suggest the introduction section may add some references from other countries about this topic such as different types of domestic work and mental health or inequity in responsibility for domestic work in the family and mental health. In the present manuscript, the author repeatedly addressed the association between unpaid domestic work and mental health from the results of Ervin’s study or their prior study in Sweden. It can be enriched with further references drawing from the current rich international bibliography or references, especially during the pandemic period.
Authors reply: Thank you for your comment. We have added references from other countries to the introduction and elaborated the introduction concerning the pandemic (please see also our reply to reviewer 1).
Materials and Methods
- As you mentioned in the manuscript, the relevant topic of this study has been published before. I suggest that this manuscript still address briefly how to design this study’s sampling and studying process or cite the previous study's reference to let readers understand thoroughly.
Authors reply: We have added some details about the survey and a reference to the previous study.
- The author needs to explain why you adopt spending 3- 10 hours per week as the reference category in this study. Is there any study recommending adopting this cut-point to analyze? This part should mention in the revised manuscript.
Authors reply: We added an explanation why 3-10 hours per week was used as the reference category, since we know from previous work that in this category the prevalence of burdensome domestic work is lowest.
Results
- Parahraph 1, lines 124-129:
In the present manuscript, rounding the number was adopted. Please let the number of percentages accord your table 1 and avoid readers getting confused.
Authors reply: We have now used decimals so that the numbers are exactly the same as in the tables. This was also done in the discussion and abstract.
Discussion
- Lines 169-170:
Rounding the number was adopted. Please let the number of percentages accord your table 1 and avoid readers getting confused.
Authors reply: See above.
- Lines 173-178:
You mentioned that “ among women, the lowest prevalence of depression was found among those who spend 11-30 hours per week in domestic work. Among men, the prevalence of self-reported diagnosed depression was highest among those who spend 0-2 hours per week in domestic work but no other statistically significant differences in depression between categories of time spent in domestic work were found”. The authors should discuss or clarify here why these results appear like this and does your study have any practical and specific policy implications in this area. Please discuss your opinion and add this important information with scientific references to the discussion part.
Authors reply: This was the result concerning self-reported depression. We have discussed the probability of reverse causality what comes to the category 0-2 hours per week and that the effect in women, even though statistically significant, was smaller than that for burdensome domestic work. Nevertheless, we have added a comment on the potential policy implications of our results.
- Paragraph 2 (page 6):
The previous references of Ervin et al. [6] and others [7] have concluded that investigating time spent in unpaid labor may however not be sufficient to assess the association between exposure to unpaid labor and mental health. However, your study still adopted time spent to examine this objective. This leads to misunderstanding results and may lead to bias. This part should discuss and mention the study's limitations.
Authors reply: We have added a short comment to the introduction why we chose to explore the association between time spent in unpaid labor and mental health as our objective. We have also commented this in the discussion. We do think that it is an important question since the review of Ervin et al. only used time in domestic work to measure exposure to domestic work, but we do not see why this would lead to bias.
- That standard instruments, such as the concept of depressive symptoms, experience domestic work as burdensome, social support, economic difficulties, and were not applied in the present study, could have led to a misclassification of information, may have affected the validity and reliability of the questionnaire, and may have caused a potential lack of comparability to the results of prior studies that used standard instruments. This part should discuss and mention the study's limitations.
Authors reply: We have extended the discussion on the validity of the measures of domestic work and mental health used with some additional references. The measures of social support and economic difficulties were the same as in the national public health survey and are commonly used in previous studies.
- This study has a lower response rate as well as 43%. Selection bias due to non-participants was inevitable. This part should discuss and mention the study's limitations.
Authors reply: We agree and have added this to the limitations. Since the limitations part became rather long, we added a subtitle for it.
- Page 7, lines 238-239:
You mentioned that “This category covers, however, a very small proportion of the adult population; 5% among women and 10% among men”. In this manuscript, I did not see any data to demonstrate this sentence. Please confirm it or add the necessary material to the revised manuscript.
Authors reply: These proportions were given in Table 1. We have added decimals so that it is easier to see that the numbers are the same.
- In this study, the experience of domestic work as burdensome was an important predictor instead of time spent hours in domestic work. The authors discussed a lot in time spent hours. I suggest the authors should focus on discussing the impact of the experience of domestic work as burdensome on mental health in the discussion part.
Authors reply: We have discussed time spent in domestic work quite a lot since this was the objective of the study. But we have extended the discussion on burdensome domestic work and the implications of these results.